# Genetic Lesions in Russian CLL Patients with the Most Common Stereotyped Antigen Receptors

**DOI:** 10.3390/genes14020532

**Published:** 2023-02-20

**Authors:** Bella V. Biderman, Ekaterina B. Likold, Nataliya A. Severina, Tatiana N. Obukhova, Andrey B. Sudarikov

**Affiliations:** National Medical Research Center for Hematology, Novy Zykovski lane 4a, Moscow 125167, Russia

**Keywords:** CLL, *IGHV*, stereotyped antigenic receptors, *TP53*, *NOTCH1*, *SF3B1*, del17p, del11q, complex karyotype

## Abstract

Chronic lymphocytic leukemia (CLL) is one of the most common B-cell malignancies in Western countries. *IGHV* mutational status is the most important prognostic factor for this disease. CLL is characterized by an extreme narrowing of the *IGHV* genes repertoire and the existence of subgroups of quasi-identical stereotyped antigenic receptors (SAR). Some of these subgroups have already been identified as independent prognostic factors for CLL. Here, we report the frequencies of *TP53*, *NOTCH1,* and *SF3B1* gene mutations and chromosomal aberrations assessed by NGS and FISH in 152 CLL patients with the most common SAR in Russia. We noted these lesions to be much more common in patients with certain SAR than average in CLL. The profile of these aberrations differs between the subgroups of SAR, despite the similarity of their structure. For most of these subgroups mutations prevailed in a single gene, except for CLL#5 with all three genes affected by mutations. It should be noted that our data concerning the mutation frequency in some SAR groups differ from that obtained previously, which could be due to the population differences between patient cohorts. The research in this area should be important for better understanding the pathogenesis of CLL and therapy optimization.

## 1. Introduction

Chronic lymphocytic leukemia (CLL) is one of the most common B-cell lymphoproliferative diseases among the adults in Russia and in Western countries. This malignancy is characterized by extreme heterogeneity—some patients have an aggressive course of the disease and a poor response to therapy, while others may not require treatment for a long time. A key role in the development of CLL belongs to the B-cell receptor and the immunoglobulin genes, coding for its main components. In the late 1990s, it was shown that the usage of immunoglobulin heavy chain variable genes (*IGHV*) in patients with CLL was substantially different from that in B-cells of healthy people [1]. In the same study it was demonstrated that *IGHV* genes were subjected to somatic hypermutation (SHM) in many patients with CLL. A few years later, it was shown that all cases of CLL were divided into two wide groups with dramatically different clinical outcomes depending on the status of SHM [2,3]. Cases with 98–100% similarity of *IGHV* gene to germinal gene agreed to be considered unmutated while cases with less than 98% identity of *IGHV* gene to the germinal were considered to be mutated. Two independent studies have shown that patients with non-mutated *IGHV* genes had an aggressive course of the disease, a short time to the first treatment, poor response to chemotherapy, and low overall survival. On the contrary, cases with mutations in the *IGHV* genes had an indolent form of the malignancy and a good response to therapy [2,3]. After more than 20 years, the proposed stratification of patients based on the mutational status of *IGHV* remains the most important and reliable approach for prognosis in CLL [4], displacing the clinical significance of other biomarkers that may change over time. In the international clinical CLL practice accessing of *IGHV* mutation status is mandatory before the initiation of treatment [5,6].

The study of the *IGHV* gene repertoire in CLL over the past 30 years has shown its significant narrowing. It was found that in a number of unrelated cases of CLL, both with mutated and unmutated *IGHV* genes, almost identical amino acid sequences of antigen binding sites (CDR3 region), later called stereotyped antigen receptors (SAR) often occurred [7,8,9,10,11]. The existence of SAR is a strong argument in favor of the influence of certain antigens on the development of CLL, since these cases were independent and widely separated geographically. More than 40% of all CLL patients can be assigned to one of SAR subgroups; 29 most common (main) SAR subgroups each include at least 60 identical unrelated cases and account for 13.5% of all CLL cases [12]. Most of the major SARs are formed with non-mutated *IGHV*—18 subgroups from 29. The amino acid motifs of SARs with unmutated *IGHV* are more conservative [12].

Several independent studies have shown that patients assigned to the same SAR subgroup exhibit similar biological features, including the same reccurent genetic lesions, gene expression profiles, epigenetic modifications, etc. [13,14,15,16,17,18]. For example, mutations in the *SF3B1* gene were very often detected in patients with CLL#2—in 45% of cases [14,19]. At the same time, according to these research groups, mutations in this gene were observed much less frequently in patients with the expression of the non-stereotype *IGHV3–21* gene—in 13.5% of cases. Mutations in the *TP53* gene, in contrast, were extremely rare in patients with the CLL#2, which was later confirmed by other researchers [13,19,20]. On the other hand, deletions and mutations in *TP53* and *NOTCH1* genes are frequently detected in patients with the most common in Russia subgroup CLL#1—16–30% [13,20,21]. As a result of these specific features, some SAR subgroups are already considered to be independent prognostic factors in CLL [20,22]. Currently, according to the recommendations of European Research Initiative in CLL (ERIC), the presence of CLL#2 and CLL#8 in patient must be reported [4]. This is due to the fact that CLL#2 is an indicator of an unfavorable prognosis even with mutated *IGHV* genes, and CLL#8 is associated with a high risk of Richter transformation [4]. On the contrary, CLL#4 subgroup is associated with a very favorable disease course [22]. Here, we present the data concerning genetic peculiarities of Russian CLL cases assigned to the most common SAR subgroups. The frequency of *TP53*, *NOTCH1,* and *SF3B1* gene mutations and recurrent chromosomal aberrations was evaluated and compared with the results of other research groups.

## 2. Materials and Methods

***Patients***. The study included DNA from 152 CLL patients admitted to the National Medical Research Center for Hematology (Moscow, Russia) between 2012 and 2022. All diagnoses were established according to the international criteria [23]. All samples were chosen from our collection of DNAs isolated from CLL patients. Selections were made based on the expression of the main SAR by tumor cells (Table 1) [24]. The minimum requirements were: 1. SAR based on the IGHV of the 1st family, and 2. the availability of material for at least 10 patients in each group. For 57 patients, DNA was obtained before treatment, for 38 in relapse after one or more lines of therapy. For the other 57 patients, clinical data were not available. The median age of the patients was 61 years (41–85), and the ratio of men and women was 88:60 (in three patients, the gender was not known). The approval of the local ethics committee has been received.

***DNA isolation*.** Lymphocytes were obtained from 5–10 mL of blood or bone marrow by the method of erythrocyte lysis [25]. DNA was isolated using standard “salting out” procedure [25]. The DNA concentration was assessed with Qubit 4 fluorometer (Thermo Fisher Scientific) following the manufacturer’s instructions.

***Analysis of IGHV-IGHD-IGHJ rearrangements*.** *IGHV* mutational status and SAR assignment were done according to the methodology recommended by ERIC [4]. To amplify clonal *IGHV* gene rearrangement leader primers [26], or the BIOMED-2 framework 1 (FWR1) primers (in rare cases when the first option failed) were used [27]. The resulting clonal product was purified using an enzyme method with Illustra ExoProStar 1-step kit (Cytiva, Marlborough, MA, USA) and was sequenced using the BigDye Terminator v1.1 kit (ThermoFisher Scientific, Waltham, MA, USA). An ABI 3130 genetic analyzer (Thermo Fisher Scientific, Waltham, MA, USA) or a Nanophore 05 (Institute for Analytical Instrumentation Russian Academy of Science, St. Petersburg, Russia) was used for resolution of sequences. The nucleotide sequences were analyzed using the open online database IMGT/V-QUEST [28]. Only productive *IGHV-IGHD-IGHJ* rearrangements were further analyzed. Assigning cases as mutated or unmutated was done using standard 98% cut-off value for homology to the closest germline gene [4]. SAR subgroups were assigned by online software ARResT/Assign Subsets [29].

***Mutation analysis.****TP53* (exons 2–11), *NOTCH1* (exon 34), and *SF3B1* (exons 14–16) gene mutations were assessed by NGS. The target gene fragments were amplified using the primers described earlier [30,31,32]. Libraries for sequencing were prepared using Nextera XT DNA Library Prep reagents (Illumina, San Diego, CA, USA). Sequencing was performed using the MiSeq Reagent Kit v2 (Illumina, San Diego, CA, USA) on the MiSeq genetic analyzer (Illumina, San Diego, CA, USA). Data filtering, deletion of service sequences, mapping of readings, and calling for variants and annotation were carried out using Trimmomatic [33], BWA [34], SAMtools [35], Vardict [36], and Annovar [37] open-source utilities. The clinical significance of the detected variants in the *TP53* gene was analyzed using UMD TP53 data and the open online tool Seshat [38]. For *NOTCH1* and *SF3B1* genes, the pathogenic value of mutations was assessed using COSMIC [39] or Franklin by Genoox [40] online databases and other published data. Variants described as (likely) pathogenic, with a variant allelic frequency (VAF) of 3% or higher were considered significant.

***Cytogenetic studies*.** Peripheral blood or bone marrow mononuclear cells for fluorescence in situ hybridization (FISH) studies were isolated on a 1.077 g/L density gradient Lympho Separation Medium solution (ISN Biomedicals, Costa Mesa, CA, USA). FISH analysis for the detection of 17p (TP53) deletion with DNA-probe XL TP53/17cen (MetaSystems, Altlussheim, Germany) was performed in 114 cases of 152. In 23 cases (in subgroups CLL#1 and CLL#6), Vysis CLL FISH Probe Kit (Abbott Molecular, Des Plaines, IL, USA) was used for the detection of deletions 11q22.3 (ATM), 13q14 (D13S319), 17p13 (TP53), and trisomy 12. A minimum of 200 interphase nuclei were examined.

Chromosome G-banding analysis (CBA) was performed in 16 cases (in subgroups CLL#1 and CLL#6). Peripheral blood cells were cultured for 72 hr in RPMI 1640 medium with 20% fetal calf serum in the presence of the immunostimulatory CpG-oligonucleotide DSP30 (2 µM; TIB MolBiol, Berlin, Germany) and interleukin 2 (200 U/mL; Sigma-Aldrich, St. Louis, MI, USA); then, colcemid (0.15 µg/mL) was added to the culture for the last 17 hr of incubation. Harvest and slide preparation were performed according to standard cytogenetic protocol. The slides were scanned for metaphases on an automated metaphase finder (Metafer slide scanning platform, MetaSystems, Altlussheim, Germany). A minimum of 20 metaphases were analyzed. Chromosomes were classified according to the International System for Human Cytogenomic Nomenclature 2016, 2020 [41,42]. Complex karyotype was noted when three or more clonal chromosome aberrations were observed (structural and/or numerical). We compared cases with the number of chromosomal abnormalities 3, 4, 5, and >5.

***Statistical analysis.*** Descriptive statistics for discrete variables include counts and frequency distributions.

## 3. Results

To assess the frequency of genetic aberrations in different SAR subgroups, we studied the DNA from 152 patients with CLL belonging to 7 main subgroups: CLL#1, #3, #5, #6, # 7H, #28A, and #99 (Table 1). CDR3 in CLL subgroups #1 and #99 have a similar motif, differ by only 1 amino acid, and are formed with 1st clan *IGHV*. CLL#28A is composed using IGHV1-2 gene, while CLL#3, #5, #6, and #7H are composed using an *IGHV1-69* gene. All these cases represent unmutated variant of CLL. The mutational status of *TP53* gene was investigated for all patients. Data concerning *NOTCH1* and *SF3B1* gene mutations were obtained for 146 (96%) and 145 (95%) patients. FISH data for 17p deletion were available for 114 patients, for other chromosome aberrations—for 31 patient in subgroups CLL#1 and CLL#6. All data are given in Table 2 for comparison.

*SF3B1* gene mutations (exons 14–16) were detected in 30 patients (20.6%), and one had 2 variants at the same time. In most of the cases (26, 87%), VAF was more than 10%. A high frequency of *SF3B1* gene mutations was noted for CLL#3 (9/22, 41%) and CLL#5 (8/21, 38%), compared to that in other subgroups: CLL#1 (3/35, 8%), CLL#6 (4/21, 20%), CLL#99 (3/22, 13.6%), and CLL#7H (3/10, 30%). No *SF3B1* gene mutations were detected in the CLL#28A subgroup. The most common variants were p.K700E (11/30, 36%) and p.G742 D (10/30, 33%), while none of them were found in the CLL#7H subgroup. Exclusively for the CLL#7H subgroup, 2 variants in the position p.I704 were observed that never appeared in other subgroups.

*NOTCH1* gene mutations (exon 34) were found in 28 patients (19%), and one had two variants. VAF was more than 10% in 24 (85%) cases. Most often they were detected in CLL#99 (8/22, 36%), CLL#5 (5/21, 24%), CLL#28A (3/14, 21%), and CLL#1 (7/36, 19%) subgroups. For CLL#3 and CLL#7H subgroups, only one case with a *NOTCH1* gene mutation was detected. A widespread mutation c.7544_7545delCT was detected in most cases (26/29, 90%). Frame-shift deletion c.7558_7561del was found in a patient with CLL#1; c.7547_7551del—in a patient with CLL#28A and c.7413_7419del—in a patient with CLL#5. Nonsense mutation p.Q2444X was found in CLL#1; p.Q2409X and p.Y2490X— were both found in CLL#99.

*TP53* gene mutations were detected in 33 of 152 patients studied (22%), while 13 patients had 2 or more mutated B-cell clones (39%), and in 29 patients (88%), VAF was more than 10. The highest frequency of *TP53* gene mutations was observed in CLL#7H (4/10, 40%), CLL#6 (8/24, 33%), CLL#1 (10/38, 26%), and CLL#5 (5/21, 24%) subgroups. On the contrary, *TP53* gene mutations were rarely seen (1–2 cases, less than 10%) in CLL#3 and CLL#28A subgroups. A total of 38 variants were identified, including 30 missense mutations, 4 deletions, 3 nonsense mutations, and 1 splicing site mutation. The most common variant was p.R175H—4 cases in CLL#1, CLL#5, CLL#7H, and CLL28A subgroups. Variants p.R110L, p.V173L, p.H193Y, p.S241F, p.G244D, p.G245S, and p.R273H were observed 2 times each.

FISH analysis revealed 17p deletions in 25 patients out of 114 (22%). In the vast majority of cases (23, 92%), the presence of 17p deletion was combined with *TP53* mutations. In 2 cases (CLL#1 and CLL#6) 17p deletions were found without *TP53* mutations. On the contrary, in 6 cases *TP53* mutations without 17p deletions were found, and for 3 cases with *TP53* mutations 17p deletion data was not available. No 17p deletions were detected by FISH in the CLL#28A subgroup. However, in another study involving chromosomal microarray analysis, a single patient with a *TP53* mutation from this subgroup was found to have also a 17p locus loss [43].

For 20 patients in the CLL#1 subgroup and 11 patients in the CLL#6 subgroup, data for other cytogenetic aberrations was available. In the CLL#1 subgroup, 11q deletion was found in 7 (39%) patients, in 3 cases, along with 13q deletion. One patient has trisomy 12, nobody has 13q deletion as the sole aberration, and 6/10 (60%) patients had complex karyotype (CK) ≥ 5 abnormalities, 3 of them with 17p deletion and 2 cases with 11q deletion. Additionally, 3/20 (15%) patients in this group had no recurrent chromosomal abnormalities. In the CLL#6 subgroup, 11q and deletion 13q deletion as the sole aberration was found only in 1 case each, trisomy 12 was not found. Furthermore, 4/6 (66%) patients had CK ≥ 5 abnormalities, 3 cases of them with 17p deletion. No recurrent chromosomal abnormalities was found in 4/11 (36%) cases.

Combined genetic mutations were observed in 11 patients. Most often, a simultaneous presence of *TP53* and *SF3B1* gene lesions was observed (6 cases, 54%). Combined *TP53*/*NOTCH1* and *NOTCH1*/*SF3B1* gene mutations were observed in 2 patients (18%). In one patient, mutations were detected in all three genes simultaneously.

## 4. Discussion

At present, when new effective targeted drugs for the treatment of CLL appear, there is no doubt about the need for reliable biomarkers for determining a more appropriate method of therapy. Currently, the main biological features associated with this disease and affecting the treatment algorithm are the *IGHV* gene mutational status and *TP53* gene lesions. Some other biomarkers are also associated with various clinical outcomes and can potentially be used for therapy selection, but they are not introduced into routine clinical practice yet. One of these biomarkers are SAR.

The data concerning recurrent genetic aberrations for a significant cohort of Russian CLL patients with the most common SAR (except for CLL#2) have been obtained. We confirm that the genetic profiles of stereotyped CLL subgroups differ significantly from CLL in general (Table 2) [20,22]. *TP53* gene lesions occur in CLL#1, CLL#5, CLL#6, and CLL#7H subgroups 2–4 times more often than in CLL in general. These gene mutations and deletions are extremely unfavorable prognostic factors in CLL. Patients with these lesions are characterized by resistance to fludarabine treatment (even small subclones can subsequently cause treatment failure), predisposition to Richter transformation, and poor response to immunochemotherapy. *TP53* gene lesions define a high-risk group in many actual prognostic models for CLL [45]. The frequency of mutations in the *NOTCH1* gene is higher than in CLL in general in subgroups CLL#1, CLL#99, CLL#28A, and CLL#5. *NOTCH1* gene mutations are associated with a poor response to anti-CD20 antibodies, resistance to fludarabine treatment, predisposition to Richter transformation, short time before the first treatment, and an unfavorable prognosis [46,47]. Subgroups CLL#3, CLL#5, CLL#6, and CLL#7H are significantly enriched with *SF3B1* gene mutations, which are often observed in combination with unfavorable chromosomal aberration del 11q, and associated with an insufficient response to immunochemotherapy and poor prognosis [48,49]. In our study, data on cytogenetic aberrations (except del 17p) were available only for the CLL#1 and CLL#6 subgroups. Patients with CLL#1 had a higher frequency of 11q deletions, characterized by an unfavorable prognosis, compared to the general sample of CLL according to the published data [20,22]. We also performed CBA for a limited number of patients in these two groups. According to the literature data, CK with three or more chromosomal aberrations is observed in 11–18% of CLL patients who do not yet require treatment, and up to 40% in relapsed/refractory CLL patients [44,50]. Currently, it has been shown that CK ≥ 3 is not an absolutely unfavorable factor in CLL, since this group is very heterogeneous and may have a different clinical course depending on the abnormality profile. At the same time, CK ≥ 5 is a marker of an extremely unfavorable prognosis, regardless of the clinical stage and mutational status of *IGHV* and *TP53* genes [44,50]. Large-scale CK studies in various SAR subgroups have not been conducted before. Our results showed an abnormally high number of patients with five or more chromosomal aberrations: 60% for CLL#1 and 66% for CLL#6 subgroups. However, our patient sample still needs to be extended to obtain more representative data.

In addition, there are significant differences in the frequency of the recurrent genetic lesions among different SAR subgroups, even for those based on the same *IGHV* genes (Table 2, Figure 1). Besides CLL#3, CLL#5, CLL#6, and CLL#7H, subgroups are based on the *IGHV1-69* gene, and the frequency of *SF3B1* gene mutations is twice as high in CLL#5 and CLL#3 subgroups comparing to CLL#6 subgroup (38–40% vs. 20%). *TP53* gene mutations and 17p deletion, on the contrary, are more common in the CLL#6 subgroup than in CLL#3 (33% vs. 9% and 28% vs. 6%, respectively). CLL#1 and CLL#99 SAR have a similar motif and differ by only one amino acid. However, *TP53* gene mutations and 17p deletion are twice as rare in the CLL#99 subgroup compared to CLL#1 (14% vs. 26% and 11% vs. 26%), while *NOTCH1* and *SF3B1* gene mutations are almost twice as common (36% vs. 19% and 14% vs. 8%). Most frequently, recurrent mutations were detected in CLL#5 and CLL#7H subgroups (in 71% and 70% of cases). The lowest mutation rate was observed in the CLL#28A subgroup (only 26% of cases). For most of the SAR subgroups studied, mutations prevailed in one gene, except for CLL#5, where mutations affected all three genes.

Comparing our data with those obtained previously from other cohorts, it should be noted that for CLL#3 subgroup, our data is in a good compliance with that published by Sutton et al. Furthermore, there is a partial match for CLL#1 subgroup in *SF3B1* and *TP53* gene mutation data and for CLL#7H subgroup—in *NOTCH1* and *SF3B1* gene mutation data [20]. However, significant differences were observed concerning all other subgroups, especially CLL#5. According to Sutton et al., this subgroup is least mutated, while we found it to be most mutated. The number of patients with different SAR included in our study was comparable to that in the study by Sutton et al. except for CLL#1 and CLL#6 subgroups [20]. It was not possible to compare our data for gene lesions for the CLL#28 subset with that from other sources due to the lack of sufficient representative data. In Russia, this SAR is almost four times more common than in European countries—0.77% vs. 0.2% [12,24]. There is a significant discrepancy between our data and the results reported by Malchikova et al. concerning the study of *TP53* gene mutations [13]. These could be explained by insufficient sample sizes, population characteristics, or accuracy issues of the techniques used (NGS vs. Sanger sequencing). One can speculate that higher frequencies of certain recurrent genetic lesions in our study might be due to a higher percentage of relapsed/treated patients. It was previously reported that chemotherapy might drive clonal evolution of the tumor and that new mutations may appear or become more abundant after treatment [51,52]. Although we do not have complete clinical data for all the patients in our sample, it should be noted, that in almost all SAR subgroups, we have patients after chemotherapy who do not have mutations, and patients who have not received treatment, but have genetic lesions. Therefore, the past chemotherapy may not be the only cause of the differences, however, it cannot be completely excluded either.

Despite the differences in the data obtained on diverse patient cohorts it should be concluded that the frequency of the recurrent genetic lesions in certain SAR subgroups is significantly higher than in the general CLL sample (Table 2, [20,22]). The association of recurrent aberrations with certain SAR may be due to the molecular features of disease development and depend on different signaling pathways. Currently, CLL#2 or CLL#8 subgroups are proven to be independent factors of an unfavorable prognosis in CLL [4,22]. The presence of the CLL#1 receptor is also associated with a short time before the therapy initiation and with unfavorable prognosis, compared to U-CLL patients with nonstereotyped B-cell receptors, based on the same *IGHV* genes [22,53,54]. CLL#1 is the most common SAR group in Russia (3.08%) [24]. It has been reported that the clinical course of the CLL#99 subgroup with a similar motif is equally unfavorable, however according to the results of Sutton and ours, the genetic profile of theese SARs is different [12,20]. According to the data from Baliakas et al., mostly SAR, based on the *IGHV1-69* gene, which is the most common in Russia, are characterized by an aggressive course of CLL [24,55]. At the same time, CLL#3 was also associated with a high risk of AIHA [56]. Subgroup CLL#4, on the contrary, predicts a favorable course of the disease [22,53,57]. Of course, all of these facts cannot be unrelated to the likelihood of adverse genetic lesions. Despite this, we do not have complete data concerning treatment outcome for the patients included in our study, and we can speculate based on our results and the trends described above that CLL#28A subgroup may associate with more favorable prognosis than the others noted here. Further clinical research on expanded SAR sample may thus provide a new prognostic approach in CLL.

## 5. Conclusions

In this study, we confirmed that the genetic profiles of stereotyped CLL subgroups differ significantly from CLL in general, and showed some genetic peculiarities of the most common SAR subgroups in Russia. Currently, in routine Russian clinical practice, only the expression of CLL#2 and CLL#8 SAR is taken into account when choosing therapy. Our results suggest that other common SAR can potentially impact CLL diagnosis, prognostication, and therapy. Further correlation studies between various SAR, recurrent genetic aberrations, and the course of the disease, on extended patient samples are very important for better understanding of CLL pathogenesis. Since new targeted drugs and cell therapy protocols are emerging, these data should be of great importance for improving CLL treatment approaches.

## Figures and Tables

**Figure 1 genes-14-00532-f001:**
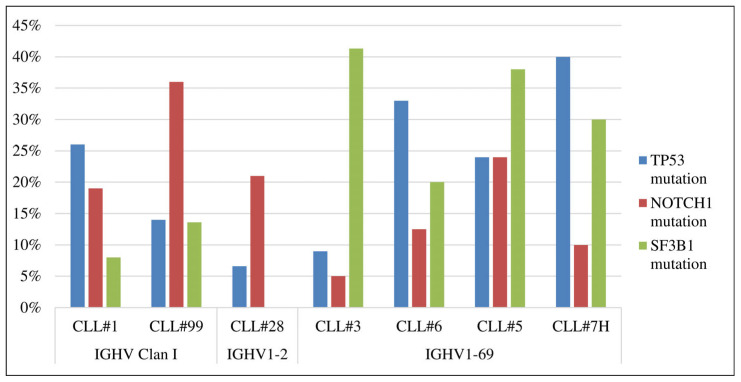
Genetic profile of the most common SAR subgroups in Russian CLL patients.

**Table 1 genes-14-00532-t001:** SAR subgroups included in the study.

Subset	IGHV Mutational Status	Subset Size *	Cases in Analysis	IGHV	Motif [12,20]
CLL#1	unmutated	3.08%	38	Clan I	ARxQWLxxxxFDY
CLL#99	unmutated	0.10%	22	Clan I	ARxQWLxxxxxFDY
CLL#28	unmutated	0.77%	15	IGHV1-2	ARx[YL]SGSYYYYYYGMDV
CLL#3	unmutated	1.45%	22	IGHV1-69	ARxxxDIVVVPAAIx[YR]YYGMDV
CLL#6	unmutated	1.16%	24	IGHV1-69	ARGGxYDY[VI]WGSYRxNDAFDI
CLL#5	unmutated	0.77%	21	IGHV1-69	ARxxxxGV[IV]xxxYYYY[GY]MDV
CLL#7H	unmutated	0.39%	10	IGHV1-69	AxxxxxxDFW[ST]GYxxxxYYYxMDV
Total		7.72%	152		

* data were also reported previously [24].

**Table 2 genes-14-00532-t002:** The frequency of genetic lesions in most common SAR subgroups of Russian CLL patients. (** in the only patient with the *TP53* mutation in this subgroup, the loss of a 17p locus fragment was subsequently detected using chromosomal micro-array analysis [43]).

	CLL#1	CLL#99	CLL#28A	CLL#3	CLL#6	CLL#5	CLL#7H	CLL in General [20,22,44]
*IGHV*	Clan I	Clan I	IGHV1-2	IGHV1-69	IGHV1-69	IGHV1-69	IGHV1-69	--
U-CLL	100%	100%	100%	100%	100%	100%	100%	50–60%
*TP53* mutation	10/38, 26%	3/22, 14%	1/15, 7%	2/22, 9%	8/24, 33%	5/21, 24%	4/10, 40%	7–10%
*NOTCH1* mutation	7/36, 19%	8/22, 36%	3/14, 21%	1/20, 5%	3/23, 13%	5/21, 24%	1/10, 10%	4–10%
*SF3B1* mutation	3/35, 8%	3/22, 13.6%	0/15, 0%	9/22, 41%	4/21, 20%	8/21, 38%	3/10, 30%	4–10%
del17p	8/31, 26%	2/18, 11%	0/6, 0% **	1/17, 6%	5/18, 28%	5/16, 31%	2/8, 25%	4%
del11q	7/20, 39%	--	--	--	1/11, 9%	--	--	11%
del13q only	0/20, 0%	--	--	--	1/11,9%	--	--	50%
tris12	1/20, 5%	--	--	--	0/11, 0%	--	--	11%
Complex kariotype	6/10, 60%	--	--	--	4/7, 66%	--	--	10–40%

## Data Availability

All the data are available from the authors upon a reasonable request.

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
