# Peer review of "Genetic Lesions in Russian CLL Patients with the Most Common Stereotyped Antigen Receptors"

_genes, 2023, doi:10.3390/genes14020532_

Round 1
Reviewer 1 Report
The authors describe the genetic peculiarities of Russian CLL patients assigned to the most common SAR subgroups. The reviewer thinks this article is very interesting for Cancer Biology, however, it needs to clear and complete in this article. Several corrections should be made.
Some comments and suggestions are given below:
1. In the Material and methods, please add more detail about how to select Russian CLL patients in a new flow chart.
2. Discussion, the author should provide more information about the effect of patients DNA before and after treatment in this study. In addition, the authors need to highlight the new findings and their connection with previous knowledge.
3. The authors are requested to briefly discuss about CLL#28 and SF3B1 mutation in the discussion and conclusions as future research and development. In addition, It would be helpful if the authors give example or scenario to support its description.
4. I suggest the language be reviewed by a native English speaker.
Author Response
Dear Reviewer,
We appreciate your valuable comments and high estimation of our work. We believe changes you have suggested should improve manuscript readability and soundness. Please find below our point to point answers to your comments.
1. In the Material and methods, please add more detail about how to select Russian CLL patients in a new flow chart.
More details added, see lines 89-92
2. Discussion, the author should provide more information about the effect of patients DNA before and after treatment in this study. In addition, the authors need to highlight the new findings and their connection with previous knowledge.
Appropriate text is added to the “Discussion” section, please see lines 304-308
3. The authors are requested to briefly discuss about CLL#28 and SF3B1 mutation in the discussion and conclusions as future research and development. In addition, It would be helpful if the authors give example or scenario to support its description.
Please see “Discussion” lines 296-299 and 329-333
4. I suggest the language be reviewed by a native English speaker.
We consulted with a native speaker about the text of the manuscript and the problematic places were rephrased in accordance with the remarks.
Best regards,
Andrey Sudarikov
Reviewer 2 Report
The study is interesting but I think that the conclusion should be improved. What is the impact of this study in choosing treatment for CLL patients? What is the impact of differences in the genetics of CLL Russian patients regarding pathogenesis and treatment?
Author Response
Dear Reviewer,
Thank you so much for your valuable comments and appreciation of our manuscript. We are confident that the manuscript should benefit from the changes that we have made in accordance with your comments. Please find below our answers to your comments with links to the relevant lines in the corrected manuscript.
-
What is the impact of this study in choosing treatment for CLL patients?
We have added appropriate statement to the “Conclusion” section (lines 337-338).
-
What is the impact of differences in the genetics of CLL Russian patients regarding pathogenesis and treatment?
This is now discussed in “Discussion” and “Conclusion” sections. Please see lines 329-333 and 339-341.
Best regards,
Andrey Sudarikov
Round 2
Reviewer 1 Report
Accepted in current form.